# Biological Control of Black Spot Disease in Cherry Tomato Caused by *Alternaria alternata* with *Bacillus velezensis* T3

**DOI:** 10.3390/foods14101700

**Published:** 2025-05-11

**Authors:** Xinmeng Wei, Qiya Yang, Dhanasekaran Solairaj, Esa Abiso Godana, Xi Zhang, Yu Li, Xiaoyong Liu, Hongyin Zhang

**Affiliations:** 1School of Life Sciences, Jiangsu University, Zhenjiang 212013, China; wxmeng120@163.com; 2School of Food and Biological Engineering, Jiangsu University, Zhenjiang 212013, China; solaibt@hotmail.com (D.S.); esa.abiso@hotmail.com (E.A.G.); xi15612806202@163.com (X.Z.); 3College of Food Science and Technology, Henan Agricultural University, Zhengzhou 450002, China; liyuliyu76@163.com (Y.L.); lxyzp01@sina.com (X.L.)

**Keywords:** *Bacillus velezensis* T3, cherry tomato, *Alternaria alternata*, biological control, disease control mechanism

## Abstract

Black spot is a major postharvest disease of cherry tomatoes, caused by *Alternaria alternata*. This causes economic losses and storage challenges, so researchers are exploring alternative methods. The biological control of fruits and vegetables using antagonistic bacteria and yeasts is currently a research hotspot. Initially, the biological control impact of *Bacillus velezensis* T3 on cherry tomato black spot was investigated. Disease defense, scavenging reactive oxygen species, and antioxidant-related enzymes were determined during different storage periods. The relative gene expressions of these enzymes were also confirmed using RT-qPCR. The results showed that *B. velezensis* T3 reduced the incidence of black spot disease in cherry tomatoes. The growth of *A. alternata* was suppressed by *B. velezensis* T3 cell-free filtrate both *in vitro* and *in vivo*. In addition, *B. velezensis* T3 induced the activities of disease resistance-related enzymes such as polyphenol oxidase (PPO), phenylalanine ammonia-lyase (PAL), β-1,3-glucanase (GLU), and chitinase (CHI), and the activities of the ROS-related enzymes superoxide dismutase (SOD), catalase (CAT), peroxidase (POD), and ascorbate peroxidase (APX), and reduced the rate of O_2_^−^ production and H_2_O_2_, and MDA content of cherry tomatoes. This approach offers a promising alternative for extending shelf life, though further studies are needed to fully characterize its effects on fruit quality.

## 1. Introduction

As favorite fruits among consumers, cherry tomatoes are rich in carotenoids, vitamins, minerals, and other nutrients [1]. Cherry tomatoes are grown all over the world and have become one of the most important crops because of their sweet taste and nutritional value [2]. However, in the process of storage and transportation, it is often attacked by pathogenic fungi, which makes the fruit deteriorate and rot, bringing great economic losses [3]. Among the several pathogenic fungi, *Alternaria alternata* (Fries) Keissler is a common postharvest pathogen of cherry tomatoes, and it causes black spot disease. Fruit tissue affected by *A. alternata* will appear as black spots with black mold on the surface [4]. Chemical fungicides are used to control postharvest diseases in cherry tomatoes. Nevertheless, certain chemicals might induce resistance in pathogen populations. Furthermore, these chemical fungicides can endure in the soil and disrupt the ecological balance [5]. There is, therefore, a need for the development of greener and safer prevention methods.

Biological control is safer and friendlier than chemical control [6]. Antagonistic microorganisms have attracted considerable interest as a promising approach for managing postharvest diseases. Common biocontrol agents include bacteria, fungi, yeasts, etc. Antagonistic yeasts have been widely used and studied in plant disease control, and several bacteria also showed promising antagonistic characteristics to control postharvest diseases of fruits and vegetables [7,8,9]. Among them, *Bacillus* species are broad-spectrum antifungal, resistant, and can survive for a longer period, and some strains have been commercially applied as biocontrol agents [10]. *Bacillus* spp. can inhibit fungal growth by competing with them for nutrients and space, and it can also produce various antifungal compounds and induce plant resistance [11,12]. A previous study found that *Bacillus amyloliquefaciens* 9001 significantly reduces the occurrence of apple ring rot [13]. He et al. proved that *Bacillus methylotrophicus* BCN2 and *Bacillus thuringiensis* BCN10 have inhibitory effects on a number of diseases, which is beneficial for the storage and preservation of loquats [14]. Research has shown the ability of *Bacillus* to produce an array of antifungal secondary metabolites, including bacteriocins, antifungal proteins, and lipopeptides [11,12]. Heo et al. demonstrated that both volatile organic compounds (VOCs) and cell-free culture filtrates from *Bacillus subtilis* GYUN-2311 exhibited broad-spectrum antimicrobial activity against various pathogens [5]. Li demonstrated that the effect on the growth of several pathogenic fungi was significant in the cell-free filtrate of *B. subtilis* LY-1 [15]. Therefore, *Bacillus* species control of postharvest diseases instead of chemical fungicides is a more promising biological control strategy.

Enhancing fruit resistance enzymes and antioxidant enzymes with *Bacillus* species boosts disease resistance. It has been reported that *Bacillus* species can increase the activity of reactive oxygen species (ROS)-related enzymes and activate the ROS system, which is beneficial for plant disease resistance [16,17,18]. Wang et al. found that treatment with *Bacillus licheniformis* HG03 increased the activity of enzymes associated with resistance to peach fruit disease, reduced the accumulation of products of membrane lipid peroxidation and favored an extension of the storage period of the fruit [19]. *B. subtilis* JK-14 induced activities of superoxide dismutase (SOD), peroxidase (POD), and catalase (CAT) in peach fruits. The induction of these three antioxidant enzymes enhanced the peach’s antioxidative defense system, which contributed to improved resistance to fungal pathogens [20]. Ji et al. found *Bacillus. licheniformis* W10 could increase defense enzyme activities and related gene expression levels in nectarine [21].

Our previous study identified a strain of *Bacillus velezensis* T3 that effectively controlled eggplant soft rot, reducing its incidence. The main antifungal substances were in cell-free filtrates [22]. However, its effectiveness against postharvest diseases in cherry tomatoes remains unknown. The present study assessed the disease-preventive potential of *B. velezensis* T3 in cherry tomatoes. The findings revealed that *B. velezensis* T3 application conferred significant protection against pathogen infection. This study further elucidated the underlying physiological mechanisms, focusing on both the direct inhibitory effects of bacterial cell-free filtrate and the induction of host defense responses. This comprehensive investigation provides valuable insights into the mechanisms by which *B. velezensis* T3 effectively controls postharvest black spot disease in cherry tomatoes.

## 2. Materials and Methods

### 2.1. Fruits

The cherry tomatoes, which were just starting to ripen, were purchased from Hanya Organic Farm in Zhenjiang, Jiangsu Province, China. Only undamaged, disease-free fruits with consistent size and coloration were used in the study. After washing the cherry tomatoes with water to get rid of any surface impurities, they were then subjected to sterilization for a duration of two minutes using a solution containing 0.1% sodium hypochlorite. The cherry tomatoes were then rinsed and left to dry.

### 2.2. Pathogen

Our research group previously isolated *A. alternata* from diseased cherry tomatoes and kept it at −80 °C. A 1 mL aliquot of fungal inoculum was aseptically transferred to sterile potato dextrose broth (PDB) and incubated for 24 h at 25 °C with continuous shaking (180 rpm). For spore production, fungal strains were grown on potato dextrose agar (PDA) at 25° C over a 7-day period. The developed cultures were subsequently processed by saline solution (0.9% sodium chloride) elution and filtration using multilayered sterile gauze to obtain spore suspensions. Spores were counted using a hemocytometer, with sterile saline adjusted to 1 × 10^5^ spores/mL.

### 2.3. Bacterial Antagonist

The *B. velezensis* T3 strain utilized in the present investigation was obtained from the Chinese General Microbiological Culture Collection Center (CGMCC) and maintained under the assigned culture collection number M2022571. Bacterial cultures grown in LB broth (16 h, 28 °C, 180 rpm) were centrifuged (8000× *g*, 5 min, 4 °C). After sequential saline (0.9% NaCl) washes, the cells were reconstituted in distilled water and normalized to the specified concentration.

### 2.4. Efficacy of B. velezensis T3 in the Control of Black Spot Disease in Cherry Tomatoes

Sterilized cherry tomatoes were punctured using a sterile perforator, creating wounds with a diameter and depth of 3 mm at the equatorial position. 10 μL of different concentrations of *B. velezensis* T3 (1 × 10^6^, 1 × 10^7^, 1 × 10^8^, 1 × 10^9^ CFU/mL) were injected into the cherry tomato wounds, while sterile saline was utilized as a control. After 2 h of inoculation with *B. velezensis* T3, 10 μL of *A. alternata* spore suspension (1 × 10^5^ spores/mL) was injected into the wound. After treatment, the fruit was stored under controlled conditions (20 °C, 95% RH). The decay incidence and lesion diameter of the pathogen across the cherry tomato wounds were measured and recorded after 3 days. Each treatment consisted of 20 cherry tomatoes, and there were three replicates for each treatment. To obtain a confirmatory result, the test was replicated twice. The following formula was used to determine the decay incidence:Decay incidence (%) = (Number of decayed fruit)/(Total number of fruits assessed) × 100

### 2.5. Inhibition of Black Spot Disease in Cherry Tomatoes by Cell-Free Filtrate of B. velezensis T3

*B. velezensis* T3 was inoculated into 50 mL of LB medium and incubated for 24 h, 48 h, and 72 h, respectively, then the liquid was collected and centrifuged at 8000 rpm for 20 min at 4 °C. Centrifugation was followed by sterile filtration of the clarified liquid fraction through a 0.22 µm membrane. The supernatant obtained after three rounds of filtration was used as the cell-free filtrate.

#### 2.5.1. In Vitro

The PDA plate was punctured to create a 5 mm hole. 50 μL of cell-free filtrate was added, along with LB medium as a control. After 2 h, 50 μL of *A. alternata* spore suspension was added. After the suspension was dried, the sample was immediately transferred to 25 °C. Colony diameters were measured after 3 days.

#### 2.5.2. In Vivo

Cherry tomatoes were punched at the midpoint, dried, and injected with 10 μL of cell-free filtrate. Controls were injected with an equal amount of LB medium. After 2 h, tomatoes were inoculated with *A. alternata* (10 μL, 1 × 10^5^ spores/mL), stored at 20 °C/95% RH. Disease progression was quantified by daily measurements of decay incidence and lesion diameter during days 3–5 of storage. Each treatment consisted of 20 cherry tomatoes with 3 replications, and whole experiments were repeated twice.

### 2.6. Effect of B. velezensis T3 on Resistance-Related Enzyme Activities of Cherry Tomatoes

From the experiment conducted in Section 2.4, we found that the concentration of the antagonistic bacteria at 1 × 10^8^ CFU/mL showed the best results. Therefore, we used this concentration for the consecutive experiments. To determine the effect of *B. velezensis* T3 on resistance-related enzyme activities of cherry tomatoes, treatments of the fruits were conducted as described above: 10 μL (1 × 10^8^ CFU/mL) of *B. velezensis* T3 was injected into the wounds. Control received equivalent sterile saline volume, maintained under identical incubation conditions (20 °C, 95% RH). Tissue samples were collected every 24 h (days 0–6). After aseptic removal of necrotic tissue, a 2 mm circumference of wound-adjacent tissue was excised for enzymatic assays.

#### 2.6.1. Determination of Polyphenol Oxidase (PPO) Activity

The PPO activity assay followed Raynaldo et al. with modifications [23]. Samples (0.5 g) were processed with 1 mL of 50 mM phosphate buffer at pH 7.8, containing 1% PVP and 1.33 mM EDTA. The enzyme solution was combined with the catechol solution in a ratio of 0.2 mL to 2.8 mL, with the catechol solution having a concentration of 50 mM. One unit (U) of PPO was defined as a 0.1 absorbance rise at 390 nm per minute per gram of cherry tomato tissue.

#### 2.6.2. Determination of Phenylalanine Ammonia-Lyase (PAL) Activity

With a minor modification, the Sun et al. approach was used to determine the PAL activity [24]. A total of 1 mL of 0.1 M borax-borate buffer (pH 8.8) was mixed with 0.5 g of cherry tomato tissue. After grinding in an ice bath, 400 µL of borax-borate buffer, 50 µL of L-phenylalanine, and 50 µL of enzyme solution were added to the tubes. The reaction was halted with 100 µL of HCl after 60 min at 37 °C. A 0.1 absorbance rise at 290 nm per minute per gram of cherry tomato tissue was used to compute one unit (U) of PAL.

#### 2.6.3. Determination of β-1,3-glucanase (GLU) Activity

The determination of GLU activity was performed according to the published procedure of Lai et al. [25]. A total of 0.5 g of cherry tomato tissue was subjected to treatment with 1 mL of phosphate buffer solution (0.1 mol/L, pH 5.7), which contained 1 mM EDTA and 5 mM β-mercaptoethanol. The tissue was then ground and subsequently centrifuged in an ice bath. A total of 880 µL of phosphate buffer solution was combined with 20 µL of enzyme solution. 100 µL of 4% kombucha polysaccharide was added to the mixture. The mixture was incubated at 40 °C, boiled, and then cooled. A total of 1.5 mL of 3,5-dinitrosalicylic acid reagent was added, heated at 100 °C, then cooled on ice. The enzyme activity unit (U) of one GLU was taken as 1 mg of glucose produced per minute of substrate degradation.

#### 2.6.4. Determination of Chitinase (CHI) Activity

Using the method described by Da Costa et al., the CHI activity was ascertained [26]. The extraction solution was identical to the one used for GLU extraction. The enzymatic assay was performed by incubating 20 μL of enzyme with 580 μL of phosphate buffer and 400 μL of 1% colloidal chitin (37 °C, 60 min). After boiling (10 min) to stop the reaction, 1.5 mL DNS was added and heated (100 °C, 10 min) for color development, followed by ice-cooling (15 min). Ultimately, the measurement of the reaction solution’s absorbance was conducted at a specific wavelength of 540 nm. The production of 1 μmol N-acetylglucosamine per minute of substrate degradation was considered as one unit (U) of CHI activity.

### 2.7. Effect of B. velezensis T3 on O_2_^−^ Production Rate, H_2_O_2_ Content, and MDA Content of Cherry Tomatoes

The cherry tomatoes were treated in the same way as described in Section 2.5. The O_2_^−^ generation rate and H_2_O_2_ content were measured using the methodology described by Zhu et al., and the results were expressed as µmol/min/g fresh weight and mg/g fresh weight, respectively [9]. Following the homogenization of 0.5 g tissue in 1 mL 10% TCA, the mixture was centrifuged (12,000 rpm, 10 min, 4 °C). The supernatant (200 μL) was then mixed with 300 μL enzyme extract and 500 μL TBA (6.7 g/L), and heated at 100 °C for 20 min. Absorbance measurements were taken at 450, 532, and 600 nm for MDA content determination.

### 2.8. Effect of B. velezensis T3 on the Activity of Antioxidant-Related Enzymes (Superoxide Dismutase (SOD), Peroxidase (POD), Catalase (CAT), and Ascorbate Peroxidase (APX)) in Cherry Tomatoes

Cherry tomatoes were treated as described in Section 2.5. The SOD enzymatic activity was determined according to the method described by Lin et al., and 50% inhibition of NBT reduction was defined as one unit (U) of SOD activity [27]. POD activity was measured following the technique of Feng et al., where an increase of 1 absorbance at 470 nm per minute per gram of cherry tomato tissue is considered as one unit (U) of POD [28]. CAT activity was determined by the method of Raynaldo et al., and one unit (U) of CAT was defined as 0.1 absorbance reduction at 240 nm per minute per gram of cherry tomato tissue [23]. APX activity was determined with reference to the method of Li et al. [29]. One unit of APX was defined as a reduction of 0.01 absorbance at 290 nm per gram of cherry tomato tissue per minute.

### 2.9. RT-qPCR for Detection of Expression Levels of Disease Resistance-Related Genes

The Bioengineering Plant RNA Extraction Kit’s instructions were followed to extract the cherry tomato RNA. HiScript II QRT SuperMix for qPCR (+gDNA wiper) was used to synthesize first-strand cDNA in accordance with the instructions, and SYBR Green SupermixTaq (Vazyme Biotech, Nanjing, China) was used for real-time fluorescence quantitative PCR (RT-qPCR). Primers used in this experiment are listed in Table 1.

### 2.10. Statistical Analysis

The statistical program SPSS version 26 (Chicago, IL, USA) was utilized to analyze the data. The Duncan’s multiple range comparison strategy was utilized for comparing means among more than two groups, while the *t*-test was implemented for comparing means between two groups. A statistical difference was deemed significant if the *p*-value was below 0.05. For every treatment, there were three independent replications.

## 3. Results

### 3.1. Efficacy of B. velezensis T3 in the Control of Black Spot Disease in Cherry Tomatoes

Figure 1A–C show the inhibition of blackspot disease in cherry tomatoes at different concentrations. All concentrations reduced the decay incidence and lesion diameter of cherry tomatoes. The decay incidence and lesion diameter of cherry tomatoes gradually decreased with increasing concentrations of *B. velezensis* T3. The decay incidence of the *B. velezensis* T3 treatment group with a concentration of 1 × 10^9^ CFU/mL was much lower compared to other concentrations, at just 10%. The lesion diameter in the 1 × 10^9^ CFU/mL *B. velezensis* T3 treatment group was 4.84 mm, which was less than in the other treatment groups and the control group. However, there was no noticeable distinction observed between the treatment groups that were administered 1 × 10^9^ and 1 × 10^8^ CFU/mL of *B. velezensis* T3. Thus, for the upcoming studies, a concentration of 1×10^8^ CFU/mL of *B. velezensis* T3 was used.

### 3.2. Inhibition of Black Spot Disease in Cherry Tomatoes by Cell-Free Filtrate of B. velezensis T3

#### 3.2.1. In Vitro

The inhibition of *A. alternata* by *B. velezensis* T3 cell-free filtrate is shown in Figure 2A,B. After 3 days of incubation, a significant reduction in *A. alternata* colony diameter was observed in the cell-free fermentation filtrate treatments compared to the control. The 72 h growth cell-free filtrate exhibited the strongest inhibitory effect on *A. alternata* colony growth among all tested samples.

#### 3.2.2. In Vivo

The inhibition effect of *B. velezensis* T3 cell-free filtrate on black spot in cherry tomatoes is shown in Figure 3A–C. The cell-free filtrates with different growth times produced significant control effects on black spot disease in cherry tomatoes. Throughout the storage period, cherry tomatoes treated with cell-free filtrates collected at different growth periods consistently showed a lower decay incidence and lesion diameter than the control group. The filtrate treatment group obtained by antagonizing bacterial growth for 72 h showed the strongest inhibition of cherry tomato black spot. This treatment group’s decay rate and lesion diameter differed considerably from those of the control group and the other treatment groups.

### 3.3. Effect of B. velezensis T3 on Resistance-Related Enzyme (PPO, PAL, GLU, and CHI) in Cherry Tomatoes

The effects of *B. velezensis* T3 on the PPO and PAL activities of cherry tomatoes are shown in Figure 4A,B. Similar patterns of PPO activity variation were observed in treated and untreated fruit throughout the storage period. In both groups, PPO activity peaked on day 1. Moreover, PPO activity in the control group continuously exhibited lower levels compared to the group treated with *B. velezensis* T3. Furthermore, there was a substantial disparity between the two groups at days 1, 2, and 4. The PAL activity of control and *B. velezensis* T3-treated cherry tomatoes also showed similar trends, fluctuating up and down throughout the storage period. PAL activity differed significantly between groups on storage days 1, 4, and 6. The control group showed lower levels of PAL activity compared to *B. velezensis* T3. GLU activity changes were similar between treated and control samples over storage time. Significant differences between both groups at days 3, 5, and 6 (Figure 4C). The control group had lower GLU activity. With the exception of days 0 and 4, CHI activity showed a marked increase in *B. velezensis* T3-treated samples compared to controls (Figure 4D). Over a period of 0–4 days, the control group exhibited a reduction followed by an increase, whereas the *B. velezensis* T3-treated group showed an increase followed by a decline. The most significant disparity between the two groups occurred on day 2. In 4–6 days, both groups showed an increase and a subsequent decrease.

### 3.4. Effect of B. velezensis T3 on O_2_^−^ Production Rate, H_2_O_2_ Content, and MDA Content of Cherry Tomatoes

According to Figure 5A, the rate of O_2_^−^ generation in cherry tomatoes increased and then dropped in both the groups treated with *B. velezensis* T3 and the control group. *B. velezensis* T3 treatment maintained significantly lower O_2_^−^ generation rates relative to untreated fruit across the entire storage period. The two groups differed significantly, except at days 0 and 4. On days 1, 2, 3, 5, and 6, the cherry tomatoes treated with *B. velezensis* T3 had a lower level of H_2_O_2_. (Figure 5B). The H_2_O_2_ content of the *B. velezensis* T3-treated group was relatively stable from 0–4 days, then decreased, and then increased on 4–6 days. The control group exhibited an initial rise followed by a decline during the first 4 days, subsequently demonstrating a progressive increase thereafter. MDA levels followed analogous trends in treated and control samples during storage (Figure 5C). The control group saw a peak in MDA content on day 1, followed by a decline. On the contrary, the *B. velezensis* T3-treated group maintained a low MDA level. Significant differences were seen between the two groups on days 1, 2, 3, and 6.

### 3.5. Effect of B. velezensis T3 on the Activity of Antioxidant-Related Enzymes (SOD, POD, CAT, and APX) in Cherry Tomatoes

Effect of *B. velezensis* T3 on SOD activity of cherry tomatoes is shown in Figure 6A. While both groups exhibited parallel SOD activity trends during storage, the *B. velezensis* T3-treated fruits consistently demonstrated elevated activity relative to controls at all time points. On days 2, 3, 4, and 6, the two groups showed significant differences. Throughout storage, both the group treated with *B. velezensis* T3 and the control group showed a gradual rise in the POD activity of cherry tomatoes (Figure 6B). *B. velezensis* T3-treated cherry tomatoes reached their highest POD on day 6, 1.53 times higher than the control. POD activity was significantly elevated in the treatment group relative to controls on days 3, 5, and 6 of storage. As shown in Figure 6C, CAT activity in treated fruit remained significantly higher than in controls at all measured time points. In particular, both groups differed significantly at days 1, 3, and 6. APX activity peaked at day 2 in all samples before decreasing, with treated fruit maintaining higher activity than controls throughout storage (Figure 6D).

### 3.6. Expression Levels of Related Genes

Figure 7 displays the gene expression levels for the enzymes PAL, APX, POD, CAT, and CHI, which are linked to defensive mechanisms in cherry tomatoes. *B. velezensis* T3 increased *PAL* expression in stored cherry tomatoes. On day 5, *APX* expression in the *B. velezensis* T3 group was 1.79 times higher than in the control. The *POD* expression in the treatment group peaked on day 4, which was 1.53 times greater than in the control group. The treatment groups’ *CAT* expression peaked at day 5, 1.7 times the control group’s level. The treatment group’s *CHI* expression peaked at day 5, 2.85 times the control group’s level.

## 4. Discussion

Black spot, also known as target spot or early blight, is a serious fungal disease mainly caused by *A. alternata*. Warm and wet growing conditions can favor the outbreak of this disease, and it causes significant losses in the cherry tomato industry in China and around the world. Various approaches have been proposed to control this disease, with chemical fungicides such as azoxystrobin considered the best method. With an increased interest in the use of organic treatments and environmental concern, other alternative approaches have been proposed to replace chemicals. The antifungal properties of *Laurus nobilis* oil are evident in its ability to restrict *A. alternata* growth, the pathogen causing black rot in cherry tomatoes [30]. The use of the antagonistic bacteria *Rahnella aquatilis* has been shown to have a promising result in reducing the susceptibility of tomato seedlings to bacterial spot disease caused by *Xanthomonas campestris pv. vesicatoria* [31].

Recently, *Bacillus* species have shown promise as bioantagonists to treat postharvest fruit and vegetable diseases. Therefore, this paper investigated the use of *B. velezensis* T3 in cherry tomato black spot to see if it could be controlled. Our results show that it is effective in controlling black spots on cherry tomatoes. As the concentration of *B. velezensis* T3 increased, the control of the decay incidence and lesion diameter of cherry tomatoes was more significant (Figure 1). Similar to observations by Bu et al., tomato treatments with higher *B. subtilis* doses showed markedly diminished disease incidence and more effective pathogen control [7]. Since lesion diameter did not differ significantly between the 1 × 10^8^ and 1 × 10^9^ CFU/mL *B. velezensis* T3 treatment groups, we concluded that 1 × 10^8^ CFU/mL was the optimum concentration for control of black spot in cherry tomatoes. Researchers have identified other antagonistic *Bacillus* species that effectively control postharvest diseases. For example, *B. amyloliquefaciens* DH4 showed potent antagonistic activity against both *Penicillium digitatum* and *P. italicum,* which are among the most destructive postharvest pathogens in citrus [32]. *B. subtilis* Y17B, a safe alternative to fungicides, has shown significant inhibitory activity against *A. alternata* [11].

It is reported that antifungal compounds produced by *Bacillus* control postharvest diseases in fruit and vegetables [33]. Lipopeptides (surfactin, bacillomycin-D, fengycin, and bacillibactin), polyketide-type antimicrobial molecules (macrolactin, bacillaene, and difficidin), and other beneficial metabolites (siderophores, bacteriocins, and volatile organic compounds (VOCs) are just a few of the many bioactive metabolites that *B. velezensis* can produce [34]. Our research demonstrated that *B. velezensis* T3 cell-free filtrate controlled black spot disease in cherry tomatoes (Figure 2 and Figure 3), confirming that antagonistic bacteria’s secondary metabolites are key to combating pathogenic fungi. From Figure 3A,B, it seems to be evident that disease was just retarded by cell-free filtrate as both parameters (decay incidence and lesion diameter) increased with time on treated samples while no significant effect of time was found in control fruits. This may be due to the fact that cell-free filtrate did not kill all pathogenic fungi but only inhibited their growth. This inhibitory effect may prolong the freshness of the fruit, reduce postharvest losses, and provide some economic benefits, which is potentially valuable for application. However, the inhibitory effect of cell-free filtrate diminished over time, suggesting that further optimization is needed for commercial application, such as increasing the concentration of antifungal substances. Several studies have documented that various *Bacillus* strains are capable of producing lipopeptides, which contribute to their biocontrol potential. Ahmad et al. demonstrated that *B. subtilis* strain Y17B effectively suppresses *A. alternata* in cherry fruit, primarily through the abundant production of secondary metabolites, including lipopeptides [11]. Guo et al. identified *B. amyloliquefaciens* M73 as an effective antagonist of postharvest pathogens in Tarocco blood oranges, with lipopeptides serving as key bioactive compounds mediating this protective effect [35]. Evidence from other studies indicates that *B. amyloliquefaciens* LZN01 produces extracellular metabolites in its cell-free supernatant capable of antagonizing *Fusarium oxysporum* [36].

Fruit resistance induced by *B. velezensis* T3 is also an important defense mechanism. PPO is an important plant defence enzyme involved in secondary metabolism, converting phenolics into quinones, which have a strong toxic effect on pathogens [37]. Through its catalytic activity in the phenylpropanoid pathway, PAL directly mediates the synthesis of important biologically active metabolites. These include plant phenols and lignins, which are important antifungal compounds that inhibit the growth of pathogens [38]. The current investigation found that the activity of PPO and PAL in cherry tomatoes dramatically enhanced following treatment with *B. velezensis* T3. (Figure 4A,B), indicating the activation of the fruit defense system. GLU and CHI function synergistically as major fungal cell wall-degrading enzymes in plant defense systems.GLU hydrolyzes β-1,3-glucan, whereas CHI hydrolyzes chitin. Such lysis of structural components compromises fungal viability and restricts pathogenic growth [39]. Treatment with *B. velezensis* T3 increased GLU and CHI activities in cherry tomatoes, improving disease resistance. (Figure 4C,D). Therefore, *B. velezensis* T3 may protect the fruit from disease by enhancing defense-related enzymes.

Plants produce large amounts of reactive oxygen species (ROS) to defend themselves against various biotic or abiotic stresses as one of the early plant defense responses. However, excessive accumulation of ROS can lead to tissue damage in fruit cells. Oxidative stress from ROS overproduction promotes membrane lipid peroxidation, which structurally and functionally destabilizes the biofilm matrix, thereby eroding cellular integrity [40]. H_2_O_2_ and O_2_^−^ are the major components of reactive oxygen species. The H_2_O_2_ and O_2_^−^ levels of treated tomatoes were significantly lower (Figure 5A,B), suggesting that *B. velezensis* T3 enhances the scavenging ability of cherry tomatoes against ROS, reduces the accumulation of ROS, and has a good biocontrol effect. MDA assesses membrane lipid peroxidation [41]. *B. velezensis* T3 application effectively maintained MDA concentrations at reduced levels compared to untreated fruit (Figure 5C). Our findings reveal the antioxidant potential of *B. velezensis* T3, as evidenced by its ability to inhibit membrane lipid peroxidation and prevent oxidative injury in cherry tomatoes.

Numerous studies have shown that antioxidant-related enzymes such as SOD, POD, CAT, and APX play an important role in scavenging ROS [42,43,44]. SOD, the key enzyme for O_2_^−^ elimination, converts mitochondrial-derived O_2_^−^ to H_2_O_2_ and O_2_, while CAT and APX constitute the principal enzymatic systems for H_2_O_2_ removal, catalytically degrading it to H_2_O and O_2_. POD is a crucial resistance-catalyzing synthase involved in plant lignin synthesis, which can help to balance H_2_O_2_ levels in the plant cell wall, thus preventing pathogen invasion. These enzymes work in cooperation to efficiently halt the buildup of reactive oxygen species and inhibit membrane lipid peroxidation [45]. In the present work, we found that *B. velezensis* T3 significantly increased the activities of SOD, POD, CAT, and APX in cherry tomatoes (Figure 6), which suggests that *B. velezensis* T3 enhances ROS scavenging capacity, activating disease resistance in cherry tomatoes. The results align with those of prior research. As demonstrated by *Bacillus licheniformis* W10 treatment, the upregulation of both antioxidant and defense-related enzymatic activities in fruit correlates with significant inhibition of *M. fructicola* infection and brown rot development [21]. *B. velezensis JZ51* increased antioxidant and defense enzyme activity, inducing disease resistance in apple fruit [38]. In addition, we randomly examined the effect of *B. velezensis* T3 on the expression of several representative resistance and antioxidant enzyme genes. *B. velezensis* T3 upregulates genes encoding the enzymes PAL, APX, POD, CAT, and CHI (Figure 7). These results provide more evidence for the theory that *B. velezensis* T3 increases the efficacy of defense-related enzymes and their expression, hence increasing disease resistance in cherry tomatoes.

## 5. Conclusions

In conclusion, this study showed that *B. velezensis* T3 had a promising efficiency to control black spot disease in cherry tomatoes. *B. velezensis* T3 can control disease through the production of antifungal substances. The cell-free filtrate of *B. velezensis* T3 had a significant inhibitory effect on the growth of *A. alternata* both *in vivo* and *in vitro*. In addition, *B. velezensis* T3 could induce host resistance and increase the activity of major disease-defense-related enzymes in cherry tomatoes. Furthermore, it can enhance the antioxidant potential of the fruit by stimulating the activity of key antioxidant enzymes. Concurrently, the activity of genes responsible for producing enzymes involved in the defense mechanism of cherry tomatoes was increased. While this study primarily evaluated the short-term biocontrol efficacy of *B. velezensis* T3 against cherry tomato black spot, our findings demonstrate its potential as an early-intervention strategy. However, further field studies are required to assess its long-term performance under natural conditions. Future research should explore optimized formulation strategies and integration with complementary approaches to enhance persistence in integrated postharvest disease management systems.

## Figures and Tables

**Figure 1 foods-14-01700-f001:**
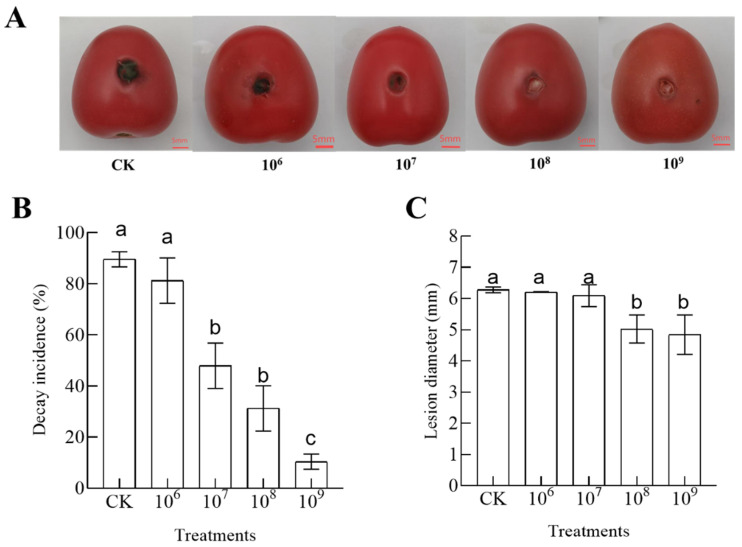
Efficacy of *B. velezensis* T3 in the control of black spot on cherry tomatoes. (**A**) control of black spot disease in cherry tomato by *B. velezensis* T3; (**B**) decay incidence; (**C**) lesion diameter. CK: the control; 10^6^–10^9^: cherry tomatoes treated with a concentration of 1 × 10^6^, 1 × 10^7^, 1 × 10^8^, and 1 × 10^9^ CFU/mL. Groups marked with dissimilar letters differ significantly (*p* < 0.05).

**Figure 2 foods-14-01700-f002:**
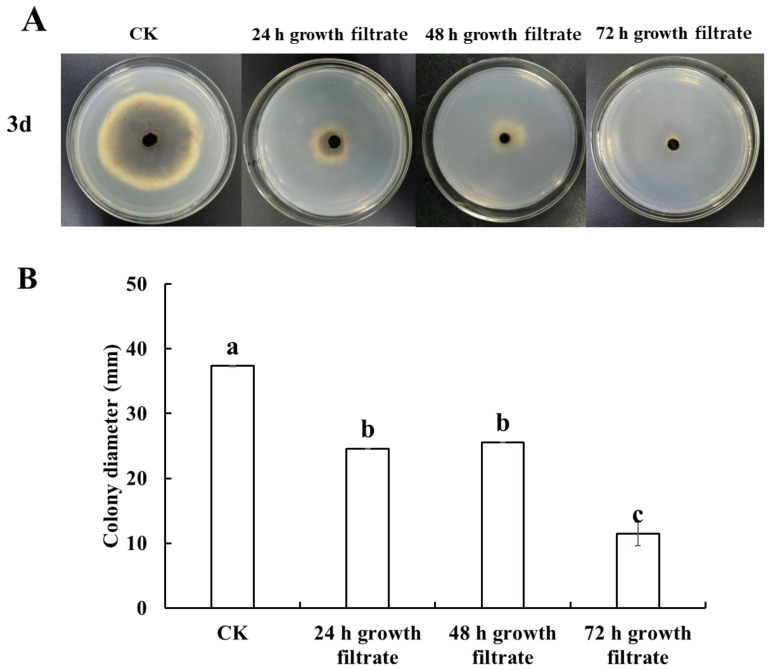
Inhibitory effect of cell-free filtrate of *B. velezensis* T3 on *A. alternata* in vitro. (**A**) photographs of colonie. (**B**) colony diameter. CK: sterile saline-treated group; 24 h growth filtrate: 24 h growth filtrate treatment group; 48 h growth filtrate: 48 h growth filtrate treatment group; 72 h growth filtrate: 72 h growth filtrate treatment group; different letters above bars indicate statistically significant differences (*p* < 0.05).

**Figure 3 foods-14-01700-f003:**
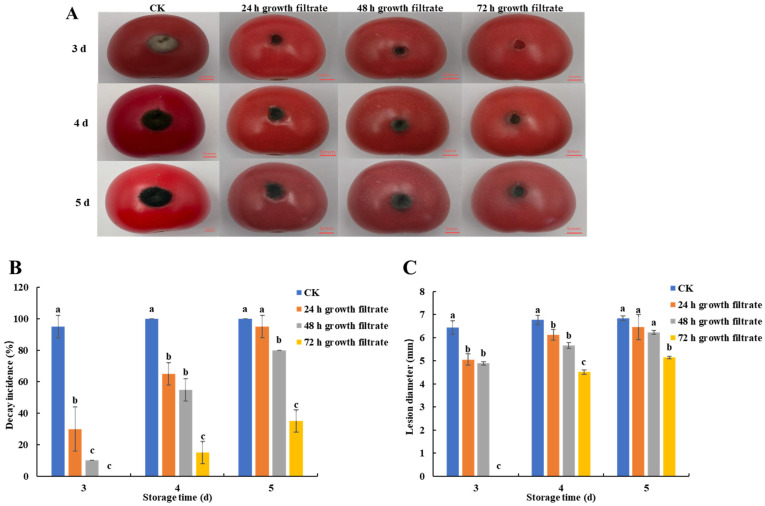
Inhibitory effect of cell-free filtrate of *B. velezensis* T3 on *A. alternata* in vivo. (**A**) the control efficacy of cell-free filtrates obtained from *B. velezensis* T3 at different growth times against black spot disease in cherry tomatoes. (**B**) decay incidence. (**C**) lesion diameter. CK: Sterile saline-treated group; 24 h growth filtrate: 24 h growth filtrate treatment group; 48 h growth filtrate: 48 h growth filtrate treatment group; 72 h growth filtrate: 72 h growth filtrate treatment group; groups marked with dissimilar letters differ significantly (*p* < 0.05).

**Figure 4 foods-14-01700-f004:**
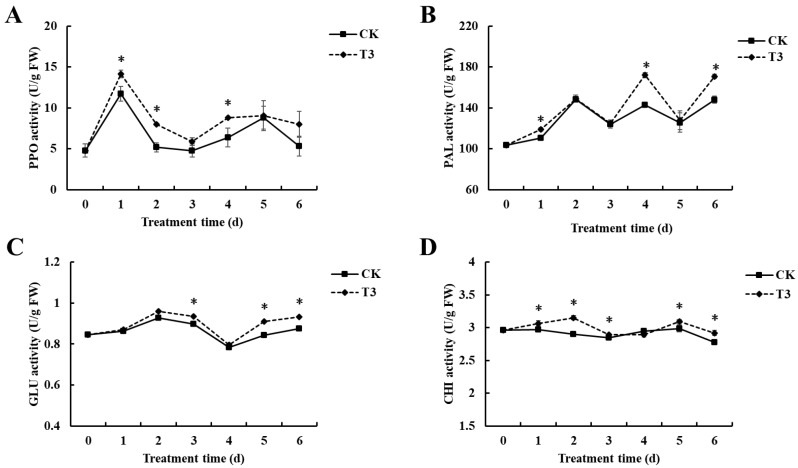
Effect of *B. velezensis* T3 on resistance-related enzyme activities in cherry tomatoes. (**A**) PPO activity. (**B**) PAL activity. (**C**) GLU activity. (**D**) CHI activity. CK: the control, sterile saline-treated cherry tomatoes; T3: Cherry tomatoes treated with *B. velezensis* T3. The asterisks indicate the statistically significant differences among treatments according to *t*-test (*p* < 0.05).

**Figure 5 foods-14-01700-f005:**
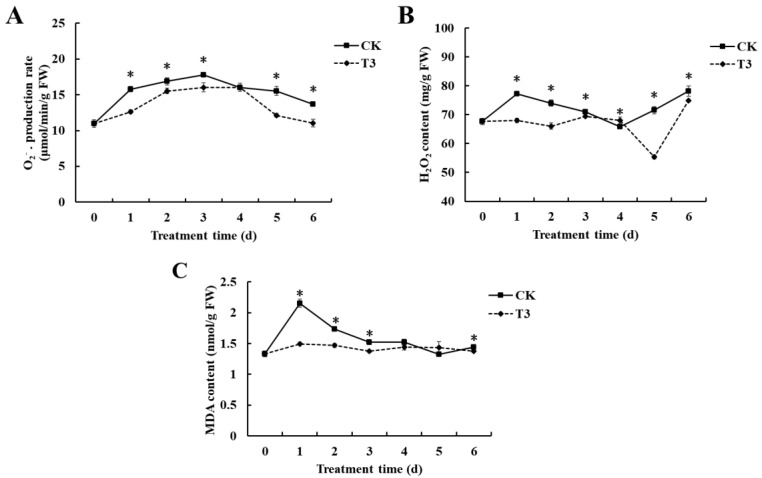
Effect of *B. velezensis* T3 on O_2_^−^ production rate, H_2_O_2_ content, and MDA content of cherry tomatoes. (**A**) O_2_^−^ production rate; (**B**) H_2_O_2_ content; (**C**) MDA content. CK: The control, sterile saline-treated cherry tomatoes;T3: Cherry tomatoes treated with *B. velezensis* T3. The asterisks indicate the statistically significant differences among treatments according to *t*-test (*p* < 0.05).

**Figure 6 foods-14-01700-f006:**
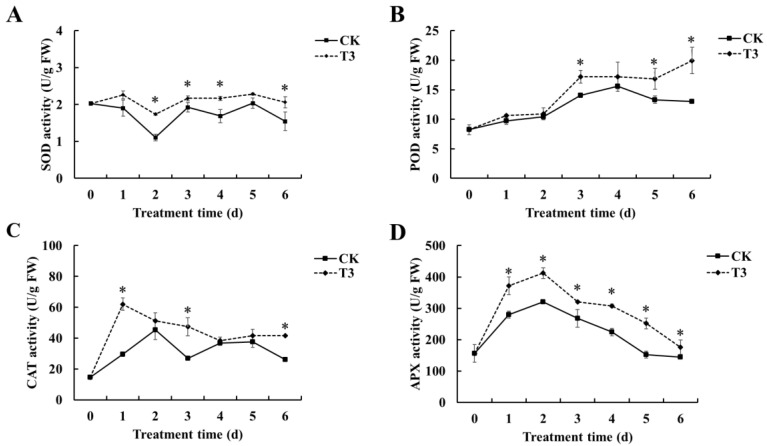
Effect of *B. velezensis* T3 on the activity of antioxidant-related enzymes in cherry tomatoes. (**A**) SOD activity; (**B**) POD activity; (**C**) CAT activity; (**D**) APX activity. CK: the control, sterile saline-treated cherry tomatoes; T3: cherry tomatoes treated with *B. velezensis* T3. The asterisks indicate the statistically significant differences among treatments according to *t*-test (*p* < 0.05).

**Figure 7 foods-14-01700-f007:**
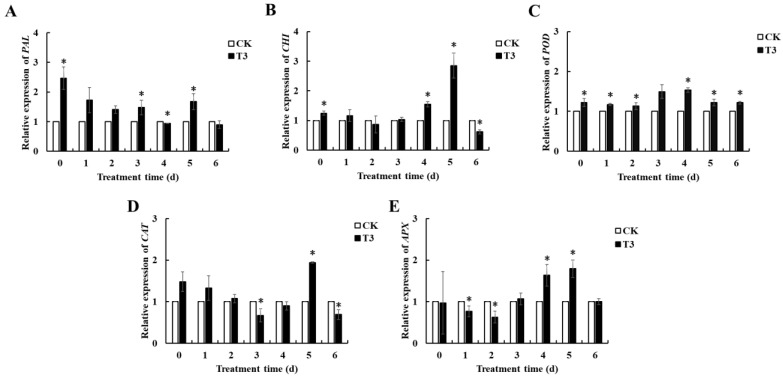
Effect of *B. velezensis* T3 treatment on the expression of disease defence-related genes in cherry tomatoes. (**A**) *PAL*. (**B**) *APX.* (**C**) *POD*. (**D**) *CAT*. (**E**) *CHI*. Treatments: control (CK) and *B. velezensis* T3). The asterisks indicate the statistically significant differences among treatments according to *t*-test (*p* < 0.05).

**Table 1 foods-14-01700-t001:** Primer sequences used for RT-qPCR.

Gene	GeneBank Number	Forward Primer (5′→3′)	Reverse Primer (3′→5′)
*Actin*	AB199316.1	acaccctgttctcctgactg	agagaaagcacagcctggat
*PAL*	Solycl0g086180.2	gcatccggtgatcttgttcc	cgaagccaaaccagaaccaa
*APX*	LC203076.1	gaggcccgaaaattcccatg	caaatgagcagcaggggaag
*POD*	NM_001247041.2	acagctcctccgaattccaa	ggaatcacgagcagcaagag
*CAT*	M37151	tgttgagggggttgtcactc	cgtgaagtccaggagcaagt
*CHI*	FJ849060.1	tggtggtagtgcaggaacat	tgtccagctcgttcgtagtt

## Data Availability

The original contributions presented in the study are included in the article, further inquiries can be directed to the corresponding author.

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
