# Peer review of "Biological Control of Black Spot Disease in Cherry Tomato Caused by Alternaria alternata with Bacillus velezensis T3"

_foods, 2025, doi:10.3390/foods14101700_

Round 1

Reviewer 1 Report

Comments and Suggestions for Authors

The manucript entitled Biological control of black spot disease of cherry tomato caused by Alternaria alternata with Bacillus velezensis T3 is a fairly complete work that addresses the ability of B. velezensis T3 to control A. alternata in tomato and describes its possible effect by activating defense mechanisms. The work is well planned both in its protocols and in the presentation of the results and only a few small details need to be highlighted and corrected

  1. Standardize the way exponents are represented in main the text and figure legends: 108 or 10^8.
  2. In Figures 2 and 3 it would be better to use the term growth to indicate how long the bacterial filtrate has been growing.
  3. It is unclear whether the filtrate culture period corresponds to the growth of velezensis or to the time it has been in the fruit prior to inoculation with the pathogen. Please describe this further.

Author Response

Dear reviewer:

We would like to thank you for your precious time and effort in providing valuable comments on our manuscript. The feedback and suggestions have helped us improve the quality and content of our manuscripts. The changes made in the revised manuscript are highlighted in blue. Here, we respond to the reviewer's comments point by point.

Comments:

The manuscript entitled Biological control of black spot disease of cherry tomato caused by Alternaria alternata with Bacillus velezensis T3 is a fairly complete work that addresses the ability of B. velezensis T3 to control A. alternata in tomato and describes its possible effect by activating defense mechanisms. The work is well planned both in its protocols and in the presentation of the results and only a few small details need to be highlighted and corrected.

Response: Dear reviewer, thank you for taking your precious time to review and evaluate our work. We appreciate your affirmation, an here are our line-by-line responses to each question.

1.Standardize the way exponents are represented in main the text and figure legends: 108 or 10^8.

Response: Dear reviewer, thank you for your kind suggestions. We have now standardized all exponential notations to superscript format (10⁸) throughout the manuscript, including main text and figure legends.

2.In Figures 2 and 3 it would be better to use the term growth to indicate how long the bacterial filtrate has been growing.

Response: Dear reviewer, thank you for your kind suggestions. As recommended, we have revised Figures 2 and 3 to consistently use the term "growth" when referring to the cultivation duration of bacterial filtrates (e.g., "24-h growth filtrate"). All related figure legends and main text descriptions have been updated.

3.It is unclear whether the filtrate culture period corresponds to the growth of B. velezensis or to the time it has been in the fruit prior to inoculation with the pathogen. Please describe this further.

Response: Dear reviewer, thank you for pointing out an important question here. The incubation period (24/48/72 h) refers specifically to the duration of in vitro growth of B. velezensis in LB medium and not in fruit tissue. We mentioned it in the Methods section of 2.5. I hope it is clear for your question.

Reviewer 2 Report

Comments and Suggestions for Authors

Abstract
Provide a concise description of the methodology, detailing the tests performed and the treatments applied to the tomatoes.
Although the authors state that “This study provides a new strategy for post‑harvest disease control in cherry tomatoes,” they should consider that such a technology may alter certain physicochemical or sensory attributes, and this possibility must be acknowledged.

Introduction
Lines 84‑97: It is not advisable to present specific objectives and methodological variables in the Introduction. I recommend merely outlining the methodological approach in brief.

Materials and Methods
Line 112 – PDB and line 114 – PDA: spell out the full name of each medium at first mention, followed by the abbreviation in parentheses.
Why were wounds made in the fruit to inoculate Bacillus velezensis T3 when evaluating the treatment effects? Under practical conditions, B. velezensis T3 would likely be applied by dipping or spraying rather than by wounding. Inoculating through injuries may have enhanced or forced its action, and the outcome could differ under conditions closer to commercial practice.

Results
For clearer interpretation of the data, I suggest integrating the discussion with the results section.

General Remarks
The English appears to be well written overall. Please review the manuscript to ensure that all scientific names are italicized (some currently are not) and to correct any remaining formatting inconsistencies.

Author Response

Dear reviewer:

We would like to thank you for your precious time and effort in providing valuable comments on our manuscript. The feedback and suggestions have helped us improve the quality and content of our manuscripts. The changes made in the revised manuscript are highlighted in blue. Here, we respond to the reviewer's comments point by point.

Comments:

Abstract

Provide a concise description of the methodology, detailing the tests performed and the treatments applied to the tomatoes.

Although the authors state that “This study provides a new strategy for post‑harvest disease control in cherry tomatoes,” they should consider that such a technology may alter certain physicochemical or sensory attributes, and this possibility must be acknowledged.

Response: Dear reviewer, thank you for your kind suggestions. We appreciate the reviewer’s insightful observation. While this study demonstrates the efficacy of the proposed strategy for postharvest disease control in cherry tomatoes, we acknowledge that the technology may influence certain physicochemical or sensory properties. Further investigations are warranted to evaluate these potential effects in detail, ensuring the approach aligns with quality standards for postharvest produce. We improved the abstract based on your suggestion.

Introduction

Lines 84‑97: It is not advisable to present specific objectives and methodological variables in the Introduction. I recommend merely outlining the methodological approach in brief.

Response: Dear reviewer, thank you for your kind suggestions. We modified the introductory section to use a broader scientific concept to describe in Line 86-88.

Materials and Methods

Line 112 – PDB and line 114 – PDA: spell out the full name of each medium at first mention, followed by the abbreviation in parentheses.

Response: Dear reviewer, thank you for your kind suggestions. The full names of PDB and PDA have been added in Line 101-104.

Why were wounds made in the fruit to inoculate Bacillus velezensis T3 when evaluating the treatment effects? Under practical conditions, B. velezensis T3 would likely be applied by dipping or spraying rather than by wounding. Inoculating through injuries may have enhanced or forced its action, and the outcome could differ under conditions closer to commercial practice.

Response: Dear reviewer, thank you for pointing out an important question here. The use of wound inoculation in this study was primarily to standardize infection conditions and ensure consistent pathogen penetration, as Alternaria alternata typically infects tomatoes through natural micro-wounds or injuries that occur during harvesting and handling. While commercial applications would indeed involve dipping or spraying, artificially wounding the fruit allowed us to precisely evaluate the biocontrol efficacy of B. velezensis T3 under controlled, high-disease-pressure conditions. In the wounds made both the antagonistic bacteria and the pathogen was applied. This is to study the maximum inhibitory effect. Most in vivo studies are conducted in this setup. However, we acknowledge that wound inoculation may intensify microbial interactions compared to uninjured surfaces, and future studies under non-wounded, practical application conditions (e.g., spray treatments) are needed to validate these findings for commercial use.

Results

For clearer interpretation of the data, I suggest integrating the discussion with the results section.

Response: Dear reviewer, thank you for your kind suggestions. As per the journal guidelines, the discussion should be written separate from the results section and we followed the same style. In the discussion we try to further interpreted the results and discuss the results obtained during the experiment. Thank you!

General Remarks

The English appears to be well written overall. Please review the manuscript to ensure that all scientific names are italicized (some currently are not) and to correct any remaining formatting inconsistencies.

Response: Dear reviewer, thank you for your kind suggestions. We revised and checked all scientific names to ensure that the formatting was italicized in Line 25,366.

Reviewer 3 Report

Comments and Suggestions for Authors
  1. Authors must include photographs of the fruits of each treatment that show what is represented in Figure 1. The photographs must have a scale in mm.
  2. Figure 3 must include photographs of the fruits that reflect what is reported in the graphs. The fruit photos must include a reference scale in mm.
  3. The graphs have different formats. The authors must standardize the graph format.
  4. Some scientific names are not in italics, please correct where necessary.
  5. The authors use "d" to refer to days, however, this causes confusion. Please, the authors should change it to the full word "days."
  6. Authors should explain and justify why some figures show 6-day follow-up data and others show 5-day follow-up data.
  7. The quality of the graphics in Figure 7 needs to be improved.
  8. Line 418: Authors must mention which metabolites (at least three) produced by these bacteria are related to fungal control. Provide evidence with bibliographic references.
  9. Line 449: What do the authors mean by saying that ROS defend plants against abiotic stress? The authors should explain this and demonstrate it with scientific evidence.

Author Response

Dear reviewer:

We would like to thank you for your precious time and effort in providing valuable comments on our manuscript. The feedback and suggestions have helped us improve the quality and content of our manuscripts. The changes made in the revised manuscript are highlighted in blue. Here, we respond to the reviewer's comments point by point.

Comments:

1.Authors must include photographs of the fruits of each treatment that show what is represented in Figure 1. The photographs must have a scale in mm.

Response: Dear reviewer, thank you for your kind suggestions. Photographs of fruit rot were added to Figure 1.

2.Figure 3 must include photographs of the fruits that reflect what is reported in the graphs. The fruit photos must include a reference scale in mm.

Response: Dear reviewer, thank you for your kind suggestions. Photographs of fruit rot were added to Figure 3.

3.The graphs have different formats. The authors must standardize the graph format.

Some scientific names are not in italics, please correct where necessary.

Response: Dear reviewer, thank you for your kind suggestions. We have standardized the format of the charts and corrected the unitalicized scientific names in Line 25,366.

4.The authors use "d" to refer to days, however, this causes confusion. Please, the authors should change it to the full word "days."

Response: Dear reviewer, thank you for your kind suggestions. We have revised the manuscript to replace all instances of "d" with "days" for unambiguous readability.

5.Authors should explain and justify why some figures show 6-day follow-up data and others show 5-day follow-up data.

Response: Dear reviewer, thank you for your kind suggestions. The difference in follow-up durations between decay observation (5 days) and enzyme activity assays (6 days) was due to the distinct experimental requirements for each measurement. For disease progression, the 5-day period was sufficient to capture clear visual symptoms of black spot rot, as decay typically becomes pronounced within this timeframe under controlled inoculation conditions. Extending decay observation beyond this period would risk excessive tissue degradation. In contrast, enzyme activity assays required an additional day (6 days) to fully characterize the dynamic biochemical responses, including the peak and decline phases of defense-related enzymes (e.g., PPO, PAL) and antioxidant systems (e.g., SOD, CAT). This extended timeframe ensured comprehensive data on the sustained induction of protective mechanisms by B. velezensis T3.

6.The quality of the graphics in Figure 7 needs to be improved.

Response: Dear reviewer, thank you for your kind suggestions. The graphic quality of Figure 7 has been improved. We have increased the resolution of the image and increased the size of the image.

7.Line 418: Authors must mention which metabolites (at least three) produced by these bacteria are related to fungal control. Provide evidence with bibliographic references.

Response: Dear reviewer, thank you for your kind suggestions. Lipopeptides (surfactin, bacillomycin-D, fengycin, and bacillibactin), polyketide-type antimicrobial molecules (macrolactin, bacillaene, and difficidin), and other beneficial metabolites (siderophores, bacteriocins, and volatile organic compounds (VOCs) are just a few of the many bioactive metabolites that B. velezensis can produce. We added this information in Line 383-387.

Rabbee, M. F., Ali, M. S., Choi, J., Hwang, B. S., Jeong, S. C., Baek, K. 2019. Bacillus velezensis: A Valuable Member of Bioactive Molecules within Plant Microbiomes. Molecules, 24(6), 1046. https://doi.org/10.3390/molecules24061046

8.Line 449: What do the authors mean by saying that ROS defend plants against abiotic stress? The authors should explain this and demonstrate it with scientific evidence.

Response: Dear reviewer, thank you for your kind suggestions. Our description here is not entirely accurate. Plants produce large amounts of reactive oxygen species to defend themselves against various biotic or abiotic stresses as one of the early plant defense responses. We have made changes in the manuscript in Line 422-423.

Reviewer 4 Report

Comments and Suggestions for Authors

I found this study interesting and relevant. The manuscript is welle written and organized. The experiment is generally well designed and performed, moreover the methodology is appropriate and well justified. Therefore, I suggest only minor revgisions and then, the pubblication of the manuscript. Comments are as follows:

Line 16. Which sp. of Alternaria is responsible of Black spot of cherry tomatoes?

Line 41. Please write the complete scientific name of Alternaria alternata, and sp.

Line 42. Which are disease symptoms on cherry tomatoes?

Lines 103-104. It is recommended to write the sentence in the passive form.

Lines 112-114. Please specify the meaning of PDB and PDA, since this is the first time they are mentioned in the text, and rewrite the sentence to more scientifically describe the procedure followed for growing the pathogen in artificial substrates.

Line 122. Please add the complete name of substrate.

Line 127. Please correct 10 L in microliters.

Lines 259-265 (Caption of Fig. 1). I suggest writing once “Cherry tomatoes treated with a concentration of”, then adding the concentrations used and in parentheses, for each concentration, the wording used in the x-axis of the graph. 

Author Response

Dear reviewer:

We would like to thank you for your precious time and effort in providing valuable comments on our manuscript. The feedback and suggestions have helped us improve the quality and content of our manuscripts. The changes made in the revised manuscript are highlighted in blue. Here, we respond to the reviewer's comments point by point.

I found this study interesting and relevant. The manuscript is welle written and organized. The experiment is generally well designed and performed, moreover the methodology is appropriate and well justified. Therefore, I suggest only minor revgisions and then, the pubblication of the manuscript. Comments are as follows:

Response: Dear reviewer, thank you for taking your precious time to review and evaluate our work. We appreciate your affirmation, and here are our line-by-line responses to each question.

Comments:

1.Line 16. Which sp. of Alternaria is responsible of Black spot of cherry tomatoes?

Response: Dear reviewer, thank you for your kind suggestions. The primary causal agent of black spot disease in cherry tomatoes is Alternaria alternata. We mentioned it in Line 41-42

2.Line 41. Please write the complete scientific name of Alternaria alternata, and sp.

Response: Dear reviewer, thank you for your kind suggestions. We have written the complete binomial name in our revised manuscript in Line 41.

3.Line 42. Which are disease symptoms on cherry tomatoes?

Response: Dear reviewer, thank you for pointing out an important question here. The symptoms of black spot disease caused by Alternaria alternata on cherry tomatoes has been included in Line 42-44.

4.Lines 103-104. It is recommended to write the sentence in the passive form.

Response: Dear reviewer, thank you for your kind suggestions. This sentence has been rewritten in Line 93-94.

5.Lines 112-114. Please specify the meaning of PDB and PDA, since this is the first time they are mentioned in the text, and rewrite the sentence to more scientifically describe the procedure followed for growing the pathogen in artificial substrates.

Response: Dear reviewer, thank you for your kind suggestions. The full names of PDB and PDA have been added in line 101-104. This sentence has been rewritten in Line 101-104.

6.Line 122. Please add the complete name of substrate.

Response: Dear reviewer, thank you for your kind suggestions. We included the complete name of the substrate in our revised manuscript in Line 111.

7.Line 127. Please correct 10 L in microliters.

Response: Dear reviewer, thank you for your kind suggestions. 10 L has been corrected to 10 μL in Line 117.

8.Lines 259-265 (Caption of Fig. 1). I suggest writing once “Cherry tomatoes treated with a concentration of”, then adding the concentrations used and in parentheses, for each concentration, the wording used in the x-axis of the graph.

Response: Dear reviewer, thank you for your kind suggestions. The graphical annotation of Figure 1 has been modified.
